# Reprograming gene expression in 'hibernating' *C. elegans* involves the IRE-1/XBP-1 pathway

Melanie Lianne Engelfriet[1], Yanwu Guo[1], Andreas Arnold[2,3], Eivind Valen[1], Rafal Ciosk[1]*

[1]Section for Biochemistry and Molecular Biology, Department of Biosciences, University of Oslo, Oslo, Norway; [2]Division of Molecular Neuroscience, Department of Biomedicine, University of Basel, Basel, Switzerland; [3]University Psychiatric Clinics, University of Basel, Basel, Switzerland

## eLife Assessment

This **useful** study reveals that as *C. elegans*, a poikilothermic ("cold-blooded") animal, adapt to cold (4°C), they display a drastic reduction in translation (assessed by polysome profiling and SUNSET). The remaining translation (by ribo-seq) correlates with mRNA levels (by RNA-seq), and the changes in gene expression at least partially require IRE-1, an established endoplasmic reticulum stress sensor. The reviewers consider the data assessing global translation and RNA expression upon cold exposure and the data demonstrating the requirement of IRE-1 to be **solid**, but the conclusion that "transcription" is the major regulatory step and "lipid changes" can be a signal for IRE-1 activation in cold adapted worms needs substantially more evidence. Overall, this study demonstrated a good correlation between translation and RNA levels and yielded an inventory of gene changes as *C. elegans* adapt to cold, and will be of general interest to researchers interested in stress response and cold adaptation.

***For correspondence:**
rafal.ciosk@ibv.uio.no

**Competing interest:** The authors declare that no competing interests exist.

**Abstract** In nature, many animals respond to cold by entering hibernation, while in clinical settings, controlled cooling is used in transplantation and emergency medicine. However, the molecular mechanisms that enable cells to survive severe cold are still not fully understood. One key aspect of cold adaptation is the global downregulation of protein synthesis. Studying it in the nematode *Caenorhabditis elegans*, we find that the translation of most mRNAs continues in the cold, albeit at a slower rate, and propose that cold-specific gene expression is regulated primarily at the transcription level. Supporting this idea, we found that the transcription of certain cold-induced genes is linked to the activation of unfolded protein response (UPR) through the conserved IRE-1/XBP-1 signaling pathway. Our findings suggest that this pathway is triggered by cold-induced perturbations in proteins and lipids within the endoplasmic reticulum, and that its activation is beneficial for cold survival.

## Introduction

One constant feature of hibernation is a global reduction of protein synthesis. This is evident from the loss of polyribosomes (also known as polysomes, which are linked to active translation) and the inhibition of key translation factors, such as eIF2α and eEF2, which are required for the initiation and elongation phases of translation, respectively (**Chen et al., 2001**; **Frerichs et al., 1998**; **Knight et al., 2000**; **van Breukelen and Martin, 2001**). Likewise, cells from non-hibernating species, including humans,

also reduce translation in response to cold (*Hofmann et al., 2012*). Despite this, many studies infer cold-related gene functions based solely on changes in transcript levels. However, given the general suppression of protein synthesis, the relative contributions of transcriptional versus translational regulation to cold-specific gene expression remain unclear.

Hibernation has been traditionally studied in non-standard animal models like squirrels, bats, or bears. Nonetheless, simpler genetic models are often advantageous in dissecting complex biological phenomena. Thus, to better understand gene expression underlying cold adaptation, we employ the nematode *C. elegans*: a rapid, morphologically simple, and genetically tractable animal model. *C. elegans* thrive in temperate climates, indicating that, in the wild, they can tolerate cold spells (*Frézal and Félix, 2015*). In the laboratory, deep cooling of *C. elegans* either leads to death when the cooling is rapid (*Habacher et al., 2016*; *Ohta et al., 2014*; *Robinson and Powell, 2016*) or to a dormant state when the cooling is more gradual (*Habacher et al., 2016*; *Ohta et al., 2014*). Studying the latter response, which in our laboratory involves shifting *C. elegans* (grown at 20 °C) for 2 hr to 10 °C and then to 4 °C for days, we observed that cold dormancy suppresses aging (*Habacher et al., 2016*). Incubating nematodes at 4 °C also elicits a diapause-like arrest (*Horikawa et al., 2024*). Moreover, some mechanisms that facilitate *C. elegans* cold survival similarly benefit cold-treated mammalian cells (*Pekec et al., 2022*). Thus, while *C. elegans* cold dormancy and mammalian hibernation are not identical, we take the liberty of also referring to the former as *C. elegans* 'hibernation'.

Here, we demonstrate that *C. elegans*, like bona fide hibernators, respond to cold by globally reducing mRNA translation. However, the residual translation of individual transcripts generally correlates with their abundance. Since transcription is typically the key determinant of steady-state mRNA levels (*Tippmann et al., 2012*), our findings suggest that cold-specific gene expression is primarily regulated at the transcriptional level. To validate this, we focused on *lips-11*, a cold-induced gene encoding a putative lipase implicated in unfolded protein response (UPR) (*Shen et al., 2005*). The UPR consists of three stress-sensing and transducing branches: IRE1, PEK1, and ATF6, all conserved in *C. elegans* (*Hetz et al., 2020*). Once activated, the UPR restores homeostasis by various means, including through remodeling transcription and translation. We find that, in hibernating *C. elegans*, cold activates the IRE-1 branch of the UPR. This activation happens in response to protein and lipid bilayer stress in the endoplasmic reticulum (ER) and results in the expression of some cold-induced genes, which appears beneficial for hibernating nematodes.

## Results

### Protein synthesis is globally reduced in hibernating *C. elegans*

In various species and cultured cells, cooling leads to a global reduction of translation (*Chen et al., 2001*; *Frerichs et al., 1998*; *Hofmann et al., 2012*; *Knight et al., 2000*; *van Breukelen and Martin, 2001*). To test if it also applies to *C. elegans*, we used a previously described cooling paradigm and sampling across different temperatures and time points (*Figure 1A*; *Habacher et al., 2016*; *Pekec et al., 2022*). We combined it with either polysome profiling or SUrface SEnsing of Translation (SUnSET). The former method separates translated mRNAs according to the number of bound ribosomes. The latter involves the incorporation of puromycin into newly synthesized peptides and subsequent detection of the incorporated puromycin by western blotting (*Arnold et al., 2014*; *Schmidt et al., 2009*). By polysome profiling, we observed in the cold a massive loss of polyribosomes (multiple ribosomes associated with mRNAs), with a concomitant increase in monosomes (*Figure 1B*). By SUnSET, we observed a stark drop in puromycin incorporation (*Figure 1C*, *Figure 1—figure supplement 1*; note that puromycin is still incorporated at 4 °C, albeit much reduced compared with 20 °C). Thus, consistent with observations in other species, severe cooling is accompanied in *C. elegans* by a global reduction of protein synthesis.

### Transcription determines cold-specific gene expression

Although global translation decreases in the cold, the translation of specific transcripts still could be regulated, i.e., activated or repressed. To examine this possibility, we combined ribosomal profiling with total RNA sequencing (*Supplementary file 1*, *Supplementary file 2*). The biggest changes in the ribosomal occupancy of mRNAs occurred when the animals were shifted from 10°C to 4°C (*Figure 2—figure supplement 1A*). Thus, most translational remodeling appears to happen during the first day

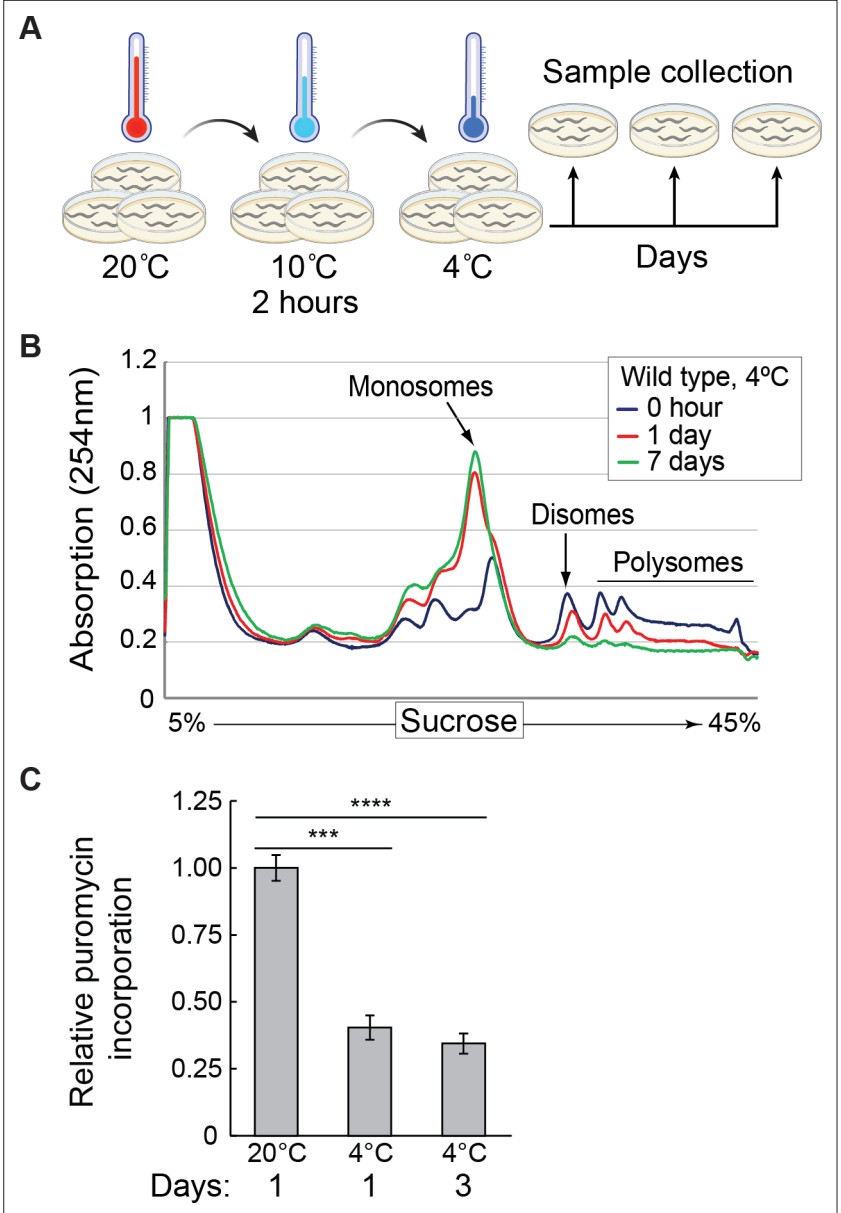

**Figure 1.** Global protein synthesis is suppressed in hibernating *C. elegans*. (**A**) Schematic of the cooling paradigm used in this study. Young adult nematodes, grown at 20 °C on multiple plates, are first adapted to the cold at 10 °C for 2 hr, and then shifted to 4 °C. At indicated time points, the animals are collected and treated in an experiment-specific manner. Created with BioRender.com. (**B**) Polysome profiles from wild-type animals were treated as shown in (**A**) and collected at 4 °C at the indicated times. Marked are the positions of mono-, di-, and polysomes. Note a strong decrease of large polysomes with a concomitant increase of monosomes in cold, which becomes more pronounced with a longer cold exposure. (**C**) Protein synthesis was evaluated with the SUrface SEnsing of Translation (SUnSET) assay in animals incubated as indicated. The quantification reflects changes in puromycin incorporation (relative to day 1 at 20 °C), normalized to actin as a loading control, and detected by western blotting. Error bars indicate the SEM of three biological replicates. Unpaired two-sided t-test was used for statistical analysis. ***p<0.001, ****p<0.0001.

The online version of this article includes the following source data and figure supplement(s) for figure 1:

**Figure supplement 1.** Protein synthesis was evaluated with the SUrface SEnsing of Translation (SUnSET) assay in animals incubated as indicated.

**Figure supplement 1—source data 1.** PDF file containing original western blots for *Figure 1—figure supplement 1*, indicating the relevant bands and treatments.

**Figure supplement 1—source data 2.** Original files depicting blots displayed in *Figure 1—figure supplement 1*.

of incubation at 4 °C. Focusing on this transition (from 10°C to 4°C on day 1), we observed a strong correlation between changes in mRNA levels and ribosomal occupancy (*Figure 2A*). This observation suggests that, overall, mRNA abundance is the main determinant of cold-specific translation (for gene categories changing the most between 10°C and 4°C day 1, see *Figure 2—figure supplement 2*). Nonetheless, we observed some exceptions, such as a group of transcripts whose translation appeared reduced in the cold with no concomitant drop in mRNA levels (red dots in *Figure 2A*), suggesting that they may be subjected to cold-specific translation repression. Intriguingly, this group includes mRNAs encoding three fatty acid desaturases (FAT-2,–3, and –4; *Supplementary file 3* and *Figure 2—figure supplement 1B*). This is surprising, considering that desaturation of membrane lipids has been thought to play an important role in cold adaptation (*Hayward et al., 2007*). However, the levels of most mRNAs correlate with the ribosomal occupancy. If the latter accurately describes translation status, the more ribosomes associated with a particular mRNA, the more protein it will yield in the cold. To strengthen this argument, we searched a public *C. elegans* depository (CGC) and identified three strains expressing GFP-fused proteins, whose ribosomal footprints either increase (*cebp-1* and *numr-1*) or not (*hsf-1*) in the cold. Monitoring their expression, we observed the expected rise in the levels of CEBP-1::EGFP and NUMR-1::EGFP, but not HSF-1::EGFP, in the cold (*Figure 2—figure supplement 3*).

Transcription is typically the key determinant of mRNA levels (*Tippmann et al., 2012*). Thus, our results suggest that cold-specific gene expression may stem from transcriptional regulation. To test this, we selected one gene, *lips-11*, whose expression increased the most during the shift from 10°C to 4°C (green dot in *Figure 2A*). To test if cold upregulates *lips-11* transcription, we generated a strain expressing a GFP reporter, whose expression depends on the endogenous *lips-11* promoter (P*lips-11*) and the *unc-54* 3'UTR (permitting unregulated expression of the attached open reading frame). We observed that the *lips-11* promoter was sufficient to upregulate GFP fluorescence (*Figure 2B*), suggesting that the expression of at least some genes in the cold is regulated at the transcriptional level.

## Cold-induced transcription of *lips-11* depends on the IRE-1/XBP-1 branch of the UPR

Intriguingly, *lips-11* belongs to genes activated by the UPR during ER stress (UPR$^{ER}$) (*Shen et al., 2005*). Indeed, we observed that the P*lips-11* GFP reporter's levels increased upon treating animals with the protein glycosylation inhibitor tunicamycin, a UPR$^{ER}$ inducer (*Figure 3—figure supplement 1*). This observation prompted us to examine if the induction of *lips-11* in the cold depends on a particular branch of the UPR$^{ER}$; IRE-1, ATF-6, or PEK-1. Indeed, we observed reduced levels of the *lips-11* reporter upon RNAi-mediated depletion of *ire-1*, but not *atf-6* or *pek-1* (*Figure 3A*). Thus, *lips-11* expression in the cold depends on the UPR$^{ER}$ signal transducer IRE-1.

In response to ER stress, the endoribonuclease domain of IRE-1 is activated and promotes the 'splicing' of *xbp-1* mRNA, which gives rise to the functional form of the XBP-1 transcription factor (*Calfon et al., 2002*). As XBP-1-independent functions of IRE-1 have been also reported, we first examined if *xbp-1* mRNA is processed in the cold. To do this, we used an *xbp-1* splicing reporter strain, wherein the *xbp-1* promoter drives the expression of a genomic *xbp-1* fragment fused to GFP, which is expressed in frame upon the processing by IRE-1 (*Ozbey et al., 2020*). Importantly, we observed a significant upregulation of XBP-1::GFP in the cold (*Figure 3B* and *Figure 3—figure supplement 2A*). Additionally, we used our ribosome profiling data to assess the translation of spliced *xbp-1* mRNA. The observed pattern of ribosomal occupancy was consistent with an increased splicing of *xbp-1* mRNA during cooling (*Figure 3—figure supplement 2B*). Finally, the cold-induced expression of the *lips-11* reporter depended on XBP-1 (*Figure 3C*). Together, these observations suggest that the upregulation of *lips-11* during cold dormancy depends on the activation of the IRE-1/XBP-1 branch of the UPR$^{ER}$.

## Hibernation specifically activates the IRE-1 branch of the UPR$^{ER}$

The activation of IRE-1 in the cold suggests that hibernating animals experience ER stress. To confirm this, we utilized a commonly used UPR$^{ER}$ reporter strain, wherein GFP expression is driven by the promoter of *hsp-4*, the *C. elegans* homolog of the ER chaperone BiP (*Calfon et al., 2002*). Indeed, the expression of this reporter increased in the cold (*Figure 4A*). By following the expression of

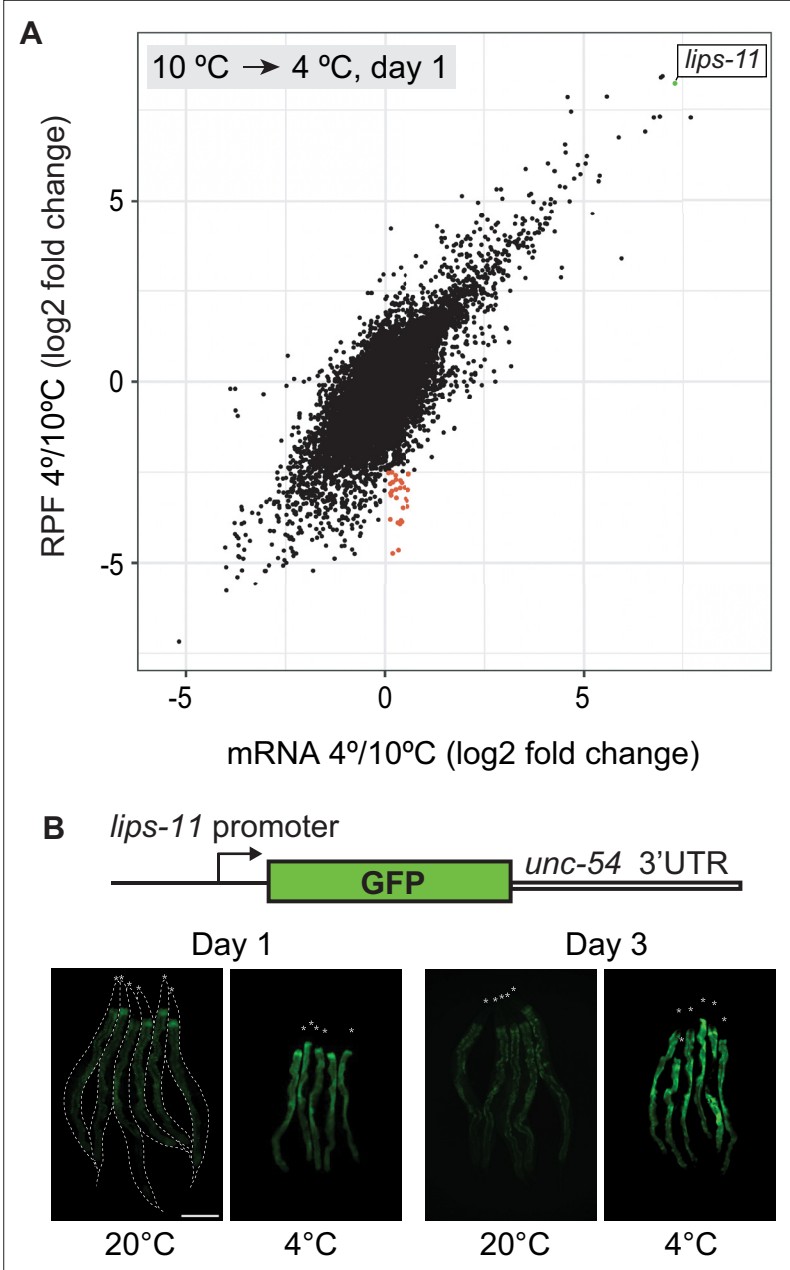

**Figure 2.** Transcription may be the main determinant of gene expression in the cold. (**A**) Changes in mRNA abundance ('mRNA' on the x-axis) and translation (ribosome protected fragment, 'RPF' on the y-axis) for all transcripts upon shifting wild-type animals from 10°C to 4°C. Each dot represents the log (base 2) fold change of a single transcript. The most upregulated transcript, *lips-11*, is indicated in green. Overall, note a strong correlation between mRNA levels and translation (Pearson correlation coefficient = 0.7244968). A small subpopulation of transcripts (red) displayed little or no change in mRNA levels but reduced association with ribosomes, suggesting specific translation repression. (**B**) Top: Diagram representing a reporter construct, wherein GFP is expressed under the control of the *lips-11* promoter and *unc-54* 3' UTR. Below are representative fluorescent micrographs, taken at the indicated conditions, of several bundled animals carrying the GFP reporter. The animals are outlined in the control panel and the heads are indicated by asterisks. The scale bar = 200 μm.

The online version of this article includes the following figure supplement(s) for figure 2:

**Figure supplement 1.** Ribosomal occupancy measured at different temperatures.

**Figure supplement 2.** Gene set enrichment analysis depicting the top 10 categories that are activated or suppressed between 10°C and 4°C on day 1 in wild-type animals.

**Figure supplement 3.** Micrographs of animals expressing EGFP fused to CEBP-1.

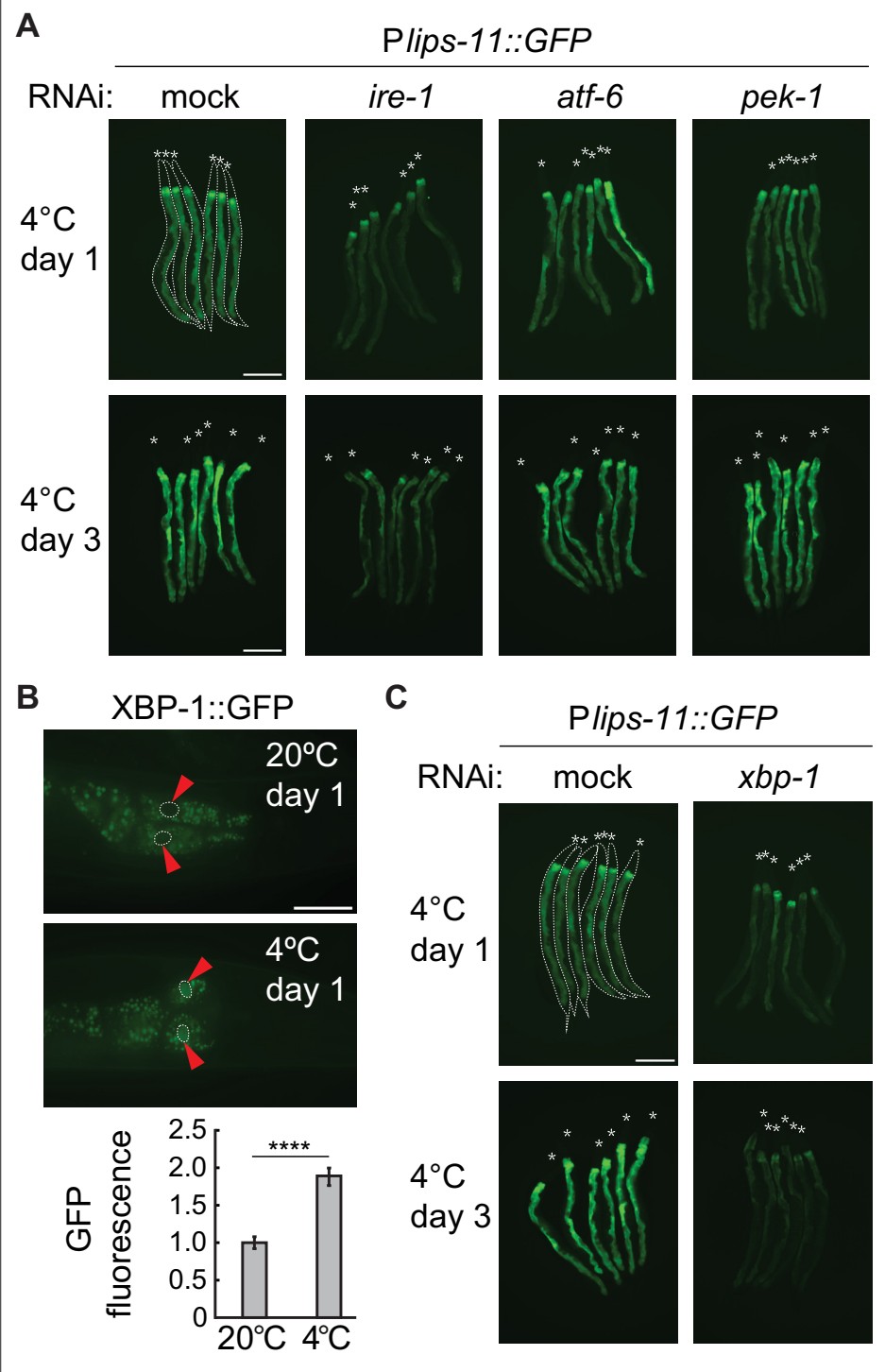

**Figure 3.** Cold-induced transcription of *lips-11* depends on the IRE-1/XBP-1 pathway. (**A**) Micrographs of several bundled animals expressing the P*lips-11::GFP* reporter, taken at the indicated conditions. Note a strong decrease in GFP expression after 1 and 3 days at 4 °C upon RNAi-mediated knockdown of *ire-1*, but not of *atf-6* or *pek-1*. The animals are outlined in the top-left panel and the heads are indicated by asterisks (**A and C**). The scale bar = 200 μm. (**B**) Top: Representative micrographs of adult animals expressing the P*xbp-1::xbp-1::GFP* splicing reporter at the indicated temperatures and time points. GFP is expressed in frame only upon removal of the IRE-1-regulated intron in *xbp-1* mRNA. Arrowheads indicate the outlined nuclei of the most anterior pair of intestinal cells. The scale bar = 40 μm. Below is the corresponding quantification of the nuclear GFP relative to 20 °C. Between two and five nuclei were analyzed per animal, in at least fifteen animals per condition. Error bars indicate

*Figure 3 continued on next page*

*Figure 3 continued*
the SEM of three biological replicates. Unpaired two-sided t-test was used for statistical analysis. ****p<0.0001. (**C**) Micrographs of mock or *xbp-1* RNAi-treated P*lips-11::GFP* reporter animals, kept for 1 or 3 days at 4 °C. The scale bar = 200 µm.

The online version of this article includes the following figure supplement(s) for figure 3:

**Figure supplement 1.** Micrographs of bundled P*lips-11::GFP* reporter animals after 6 hr of treatment with DMSO (mock) or tunicamycin at 20 °C.

**Figure supplement 2.** *XBP-1* splicing at different temperatures.

---

mitochondrial chaperones *hsp-6* (homolog of HSP70) and *hsp-60* (homolog of HSP60), we additionally examined if cold elicits UPR in the mitochondria (UPR^mito) (*Yoneda et al., 2004*). However, by contrast to *hsp-4*, neither *hsp-6* nor *hsp-60* were induced in the cold (*Figure 4—figure supplement 1*). Thus, hibernating animals appear to experience stress in the ER but not mitochondria.

While we showed that *lips-11* upregulation is mediated by the IRE-1/XBP-1 pathway, other branches of the UPR^ER could be activated in the cold. Testing this possibility, we examined the expression of select UPR^ER target genes regulated through the IRE-1 (*dnj-27*, *srp-7*, *C36B7.6*), the ATF-6 (*cht-1*, *ZC168.2*), or the PEK-1 (*cbp-3*, *R02D3.8*) pathway (*Shen et al., 2005*). Among these transcripts, four displayed increased abundance on day 3 in the cold, relative to the corresponding day 3 animals at 20 °C (*Figure 4B*). Notably, 3/4 of them are known targets of the IRE-1 pathway. Further analysis confirmed that the endogenous transcript levels of these IRE-1 responsive genes (including *lips-11*) increased in abundance as early as 1 d in the cold and continued to accumulate after 3 d, relative to reference animals collected at 20 °C immediately before cooling (day 0) (*Figure 4—figure supplement 2*). Importantly, the upregulation of these transcripts in the cold declined in the loss-of-function *ire-1(ok799)* mutants, indicating that their upregulation in the cold depends on IRE-1 (*Figure 4—figure supplement 2*). Together, these findings suggest that cooling triggers the UPR^ER, resulting in IRE-1-dependent gene expression.

## The cold-induced UPR^ER stems from both protein misfolding and lipid disequilibrium

ER stress and subsequent activation of the UPR^ER are typically associated with misfolded proteins accumulating in the ER lumen. To test if cold aggravates protein misfolding, we employed another reporter strain, wherein YFP is fused to a mutated form of the *C. elegans* cathepsin L-like protease (CPL-1^W32A, Y35A), which does not fold properly and accumulates during ER stress (*Efstathiou et al., 2022*). We found that the levels of misfolded CPL-1 modestly increased after 1 d in the cold but then returned to basal levels after a longer cold exposure (*Figure 5*). Thus, disturbed ER proteostasis may, at least transiently, trigger the activation of UPR^ER in hibernating animals.

Apart from protein misfolding, lipid disequilibrium is also known to trigger UPR^ER (*Halbleib et al., 2017*; *Tam et al., 2018*; *Volmer et al., 2013*). The IRE-1 pathway is specifically activated when the ratio between phosphatidylcholine (PC) and phosphatidylethanolamine (PE) decreases, or when the levels of unsaturated fatty acids become insufficient (*Ariyama et al., 2010*; *Hou et al., 2014*; *Thibault et al., 2012*). In the latter case, reduced content of unsaturated fatty acids (FAs) results in a decreased ER membrane fluidity, which is thought to increase the oligomerization and thus the activation of IRE-1 (*Halbleib et al., 2017*). Thus, we asked if dietary supplementation of unsaturated FAs or choline (crucial for PC synthesis) affects the IRE-1 activity in the cold. We first validated the effectiveness of FA supplementation by confirming a previous report, that adding unsaturated FAs prevents the *hsp-4* induction in *fat-6* RNAi-depleted animals (*Hou et al., 2014*; *Figure 6—figure supplement 1*). Following a similar FA supplementation procedure, we observed no reduction in the expression of the *lips-11* reporter in the cold (*Figure 6—figure supplement 2*). Thus, insufficient lipid desaturation does not seem to be the main trigger of UPR^ER in the cold. By contrast, supplementing the diet with choline reduced the expression of both *lips-11* and *hsp-4* reporters in hibernating animals (*Figure 6A–B*). Moreover, we found that choline supplementation was beneficial for cold survival (*Figure 6C*). Together, these experiments suggest that cold-induced activation of UPR^ER could be triggered by sensing disruptions in both protein and lipid homeostasis, with the latter related to insufficient levels of PC rather than unsaturated FAs.

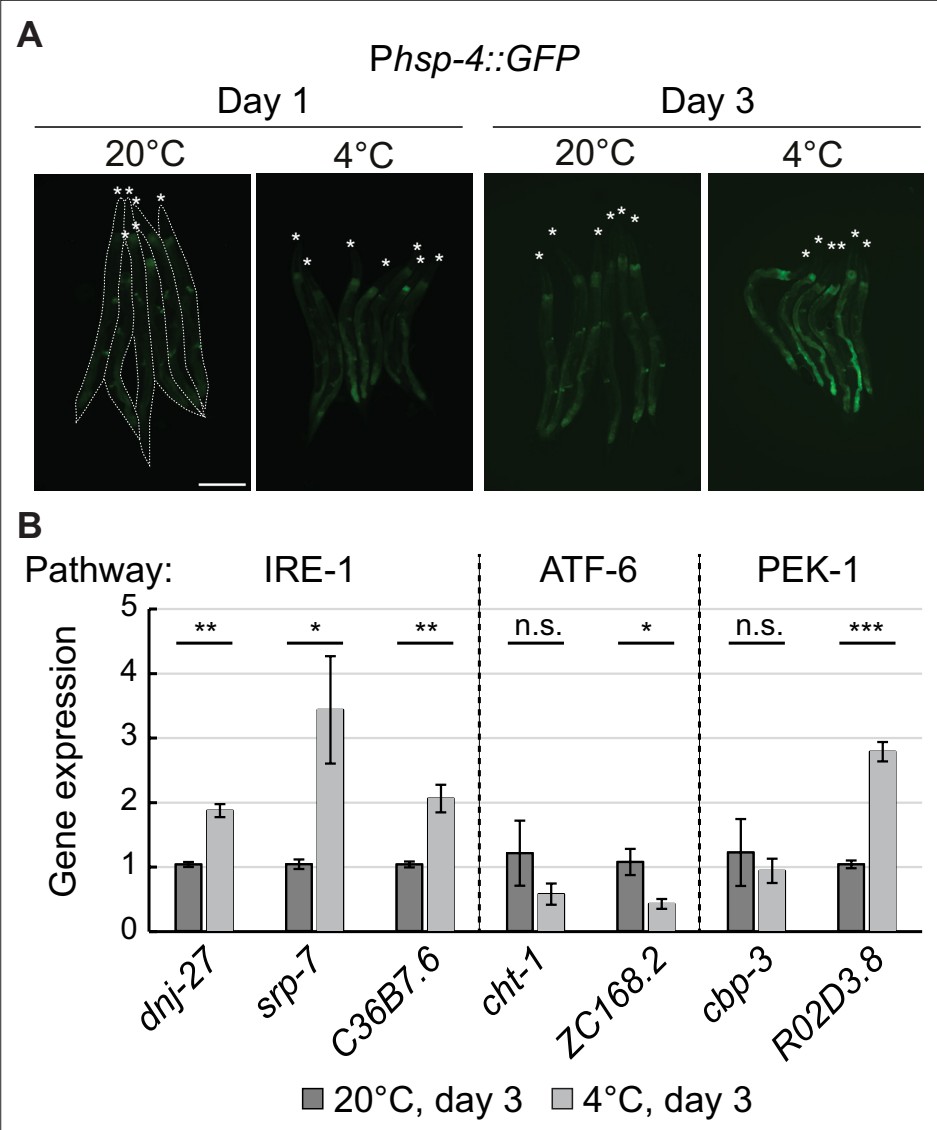

**Figure 4.** Cold specifically activates UPR[ER] through IRE-1. (**A**) Micrographs of several bundled animals carrying the P*hsp-4::GFP* reporter, taken at the indicated temperature and time. The animals are outlined in the left and the heads are indicated by asterisks. The scale bar = 200 µm. (**B**) RT-qPCR analysis of changes in mRNA levels for known UPR[ER] target genes after 3 days at 4 °C, relative to 3 days at 20 °C. Dashed lines separate genes that are regulated by different pathways (IRE-1, ATF-6, and PEK-1). Note that the mRNA levels of all IRE-1 responsive genes are significantly upregulated at 4 °C. Error bars indicate the SEM of three biological replicates. Unpaired two-sided t-test was used for statistical analysis. ns: p>0.05, *p<0.05, **p<0.01, ***p<0.001.

The online version of this article includes the following figure supplement(s) for figure 4:

**Figure supplement 1.** Mitochondrial UPR monitored as indicated.

**Figure supplement 2.** RT-qPCR analysis of changes in mRNA levels for known IRE-1 target genes in wild-type and *ire-1(ok799)* animals after 1 or 3 d at 20 °C or 4 °C, relative to wild-type animals at 20 °C day 0 (harvested immediately prior to cold adaptation at 10 °C).

## IRE-1 is important for a robust cold survival

If activating IRE-1 signaling is important during cooling, then inhibiting this pathway may be expected to impair cold survival. At standard temperature, the loss-of-function *ire-1(ok799)* mutants appear superficially wild-type. In the cold, however, these mutants displayed a modest but significant impairment of cold survival (*Figure 7A*). Thus, the IRE-1 signaling improves cold survival, presumably by activating its downstream targets. One obvious candidate is LIPS-11, but we found that its RNAi-mediated

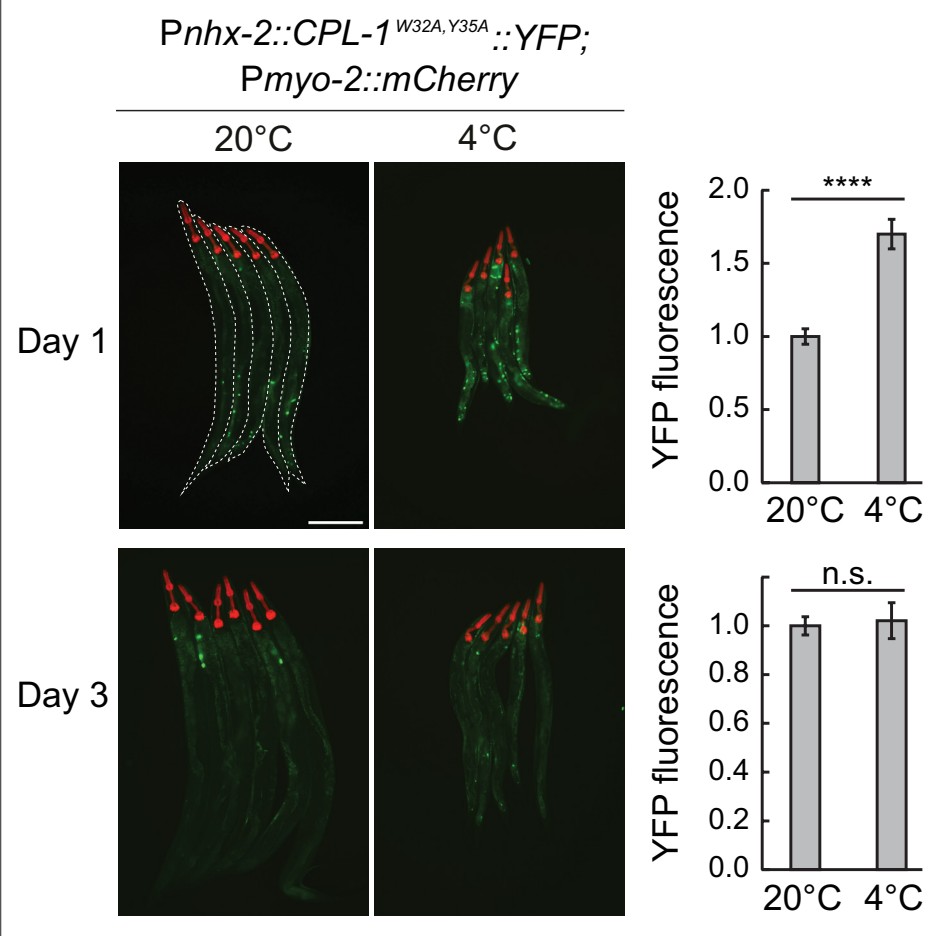

**Figure 5.** Misfolded protein levels increase transiently during cold exposure. Left: representative micrographs, taken at the indicated conditions, of bundled animals carrying the CPL-1^W32A,Y35A::YFP misfolding reporter. The head of each animal is highlighted by the pharyngeal expression of mCherry, driven by the *myo-2* promoter. Animals are outlined in the top-left panel. The scale bar = 200 µm. Right: the corresponding quantification of YFP fluorescence at 4 °C after 1 or 3 days, relative to 20 °C. The YFP fluorescence was analyzed from whole animals, with a minimum of 38 animals per condition. Error bars indicate the SEM of three biological replicates. Unpaired two-sided t-test was used for statistical analysis. ns: p>0.05, ****p<0.0001.

depletion had no obvious impact on cold survival, suggesting that LIPS-11 either plays no essential role in cold survival or functions redundantly with other IRE-1/XBP-1 targets.

Several previous studies reported IRE-1-dependent genes activated in response to protein misfolding (UPR^PT; *Shen et al., 2005*), lipid bilayer stress (UPR^LBS; *Koh et al., 2018*), or cold when using a different cooling paradigm (*Dudkevich et al., 2022*). To examine if these genes were also activated by cold treatment in our study, we extracted them from the publications (*Supplementary file 4*) and examined their overlap with the cold-induced transcripts from *Figure 2A*. Curiously, we observed little overlap between the different gene sets (*Figure 7—figure supplement 1* and *Supplementary file 5*). The UPR^PT and the UPR^LBS are already known to regulate largely distinct targets (*Koh et al., 2018*). Also, the limited overlap between those genes and IRE-1-dependent genes from Dudkevich et al. suggests that IRE-1-mediated gene expression is largely context-dependent, which could also apply to IRE-1-dependent gene expression reported here.

Summarizing, our findings suggest that cold-induced protein and lipid stress in the ER specifically induces the IRE-1 branch of the UPR. Following the IRE-1-mediated processing of *xbp-1* mRNA, the XBP-1 transcription factor activates its target genes. These include *lips-11*, but additional targets seem necessary to explain how IRE-1 benefits cold survival (*Figure 7B*). This model does not rule out additional players. For example, hibernating animals are sensitive to reactive oxygen species (ROS)

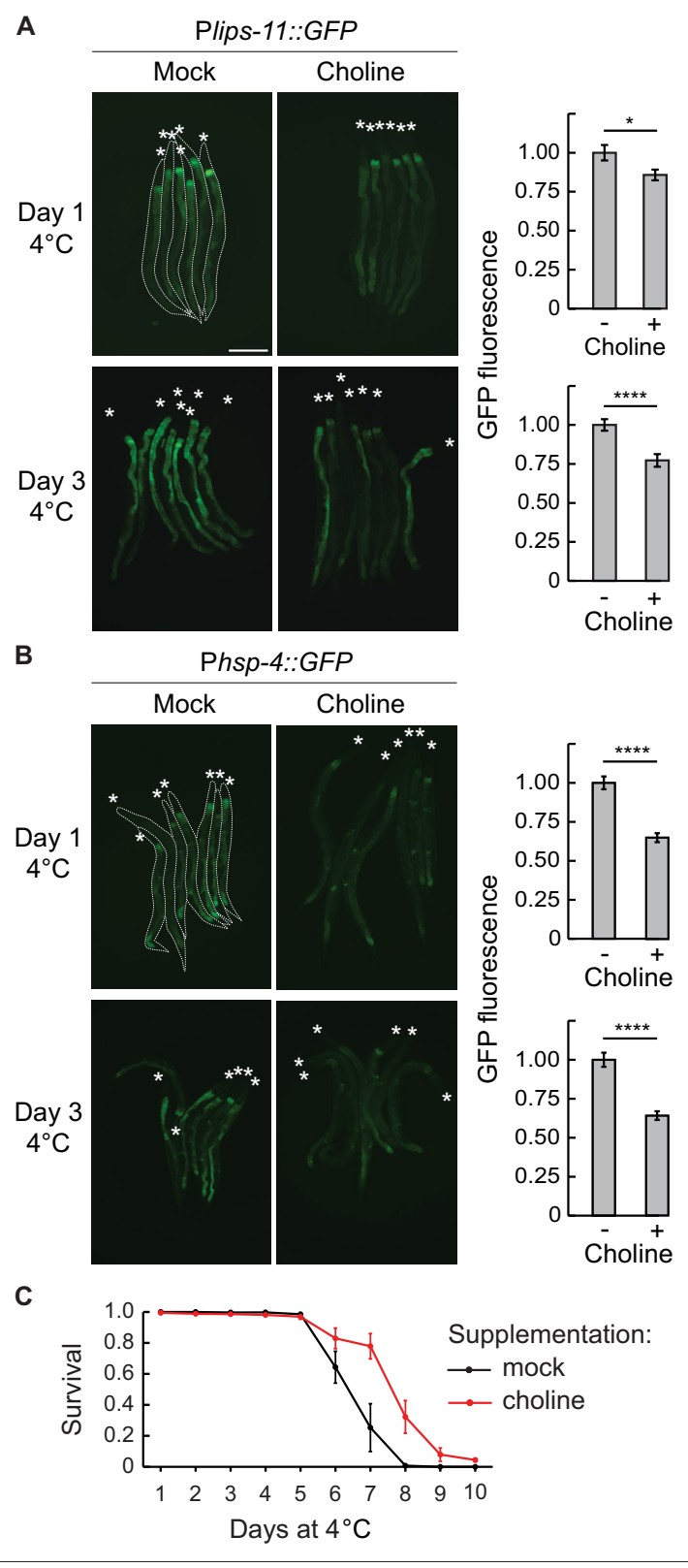

**Figure 6.** Choline supplementation suppresses UPR$^{ER}$ and improves cold survival. (**A**) Left: micrographs taken after 1 or 3 days at 4 °C of bundled P*lips-11::GFP* reporter animals on a mock or 50 mM choline supplemented diet. Animals are outlined in the top-left panel and the heads are indicated with asterisks (**A and B**). The scale bar = 200 μm. Right: the corresponding quantification of intestinal GFP fluorescence after 1 or 3 days at 4 °C

*Figure 6 continued on next page*

*Figure 6 continued*

on a choline-supplemented diet, relative to a mock diet. GFP fluorescence was analyzed from a minimum of 38 animals per condition. Error bars indicate the SEM of three biological replicates. Unpaired two-sided t-test was used for statistical analysis. *p<0.05, ****p<0.0001. (**B**) Left: Micrographs taken after 1 or 3 days at 4 °C of bundled P*hsp-4::GFP* reporter animals on a mock or 50 mM choline supplemented diet. The scale bar = 200 μm. Right: the corresponding quantification of intestinal GFP fluorescence after 1 or 3 days at 4 °C on a choline supplemented diet, relative to a mock diet. GFP fluorescence was analyzed from a minimum of 28 animals per condition. Error bars indicate the SEM of three biological replicates. Unpaired two-sided t-test was used for statistical analysis. ****p<0.0001. (**C**) Survival of wild-type animals on a mock or choline-supplemented diet. Error bars indicate the SEM of three biological replicates. A minimum of 200 animals were scored per time point. Wilcoxon signed-rank test was used for statistical analysis; p=0.02.

The online version of this article includes the following figure supplement(s) for figure 6:

**Figure supplement 1.** Micrographs of bundled animals expressing the P*hsp-4::GFP* reporter were subjected to either mock or *fat-6* RNAi and supplemented with either 0.1% tergitol (mock), 0.8 mM oleic acid, or 0.8 mM linoleic acid.

**Figure supplement 2.** Dietary supplementation of fatty acids.

---

(*Pekec et al., 2022*), and ROS can activate IRE-1-dependent antioxidant response mediated by the transcription factor SKN-1/Nrf2 (*Hourihan et al., 2016*). Whether this pathway is activated in the cold remains to be tested. Similarly, additional pathways may work alongside IRE-1 to regulate gene expression in hibernating animals. Intriguingly, we noticed that *zip-10* mRNA, encoding a transcription factor promoting organismal death during cold shock (*Jiang et al., 2018*), also goes up during cold dormancy. However, since the animals survive unabated, future studies will clarify if it plays a role in hibernating nematodes.

## Discussion

Consistent with findings from other models, our results show that also *C. elegans* respond to severe cold by globally decreasing protein synthesis. Since translation is one of the most energy-consuming biological processes, this reduction likely helps preserve cellular energy reserves. Additionally, translational regulation is reversible, which may allow for the rapid restoration of protein synthesis once animals are returned to temperatures conductive to growth and development. In poikilotherms, such as *C. elegans*, whose body temperature fluctuates with the environment, reducing global translation presumably helps the animals survive until temperatures rise again. The same applies to homeotherms capable of temporal heterothermy, meaning animals that adjust their body temperature in a circadian or seasonal circle. In other homeotherms, including humans, a global reduction of protein synthesis may serve a protective role in surviving accidental hypothermia or aiding the repair of peripheral cold injuries affecting extremities and exposed skin.

Despite the general reduction of protein synthesis during cold exposure, our results suggest that most available mRNAs are translated, albeit at a slower rate. This implies that cold-specific gene expression could be regulated through transcription or mRNA stability. In the latter case, the enrichment of specific mRNAs could result either from the degradation of specific transcripts at 20 °C (allowing them to accumulate in the cold) or from the selective degradation of some transcripts in the cold, making others relatively more abundant. While we cannot rule out this possibility for certain transcripts, we favor a simpler model in which cold-specific gene is generally regulated at the transcriptional level. Supporting this hypothesis, our findings indicate that the IRE-1/XBP-1 pathway induces transcription of at least some genes in hibernating animals.

Curiously, a different cooling protocol (shifting animals from 15°C to 2°C) also activates IRE-1 signaling, but in this case, it occurs independently of *xbp-1* processing (*Dudkevich et al., 2022*). In this case, IRE-1 activation is observed in neurons but remodels lipid metabolism in other tissues, likely to balance saturated and unsaturated FAs. Supporting this idea, dietary supplementation with unsaturated FAs bypasses the need for IRE-1 activation. This aligns with previous findings showing that desaturated FAs are crucial for survival at low but physiologically tolerable temperatures like 15 °C (*Svensk et al., 2013*). In contrast, the IRE-1 signaling described in our study involves *xbp-1* processing, and dietary supplementation with unsaturated FAs does not prevent IRE-1 activation. A

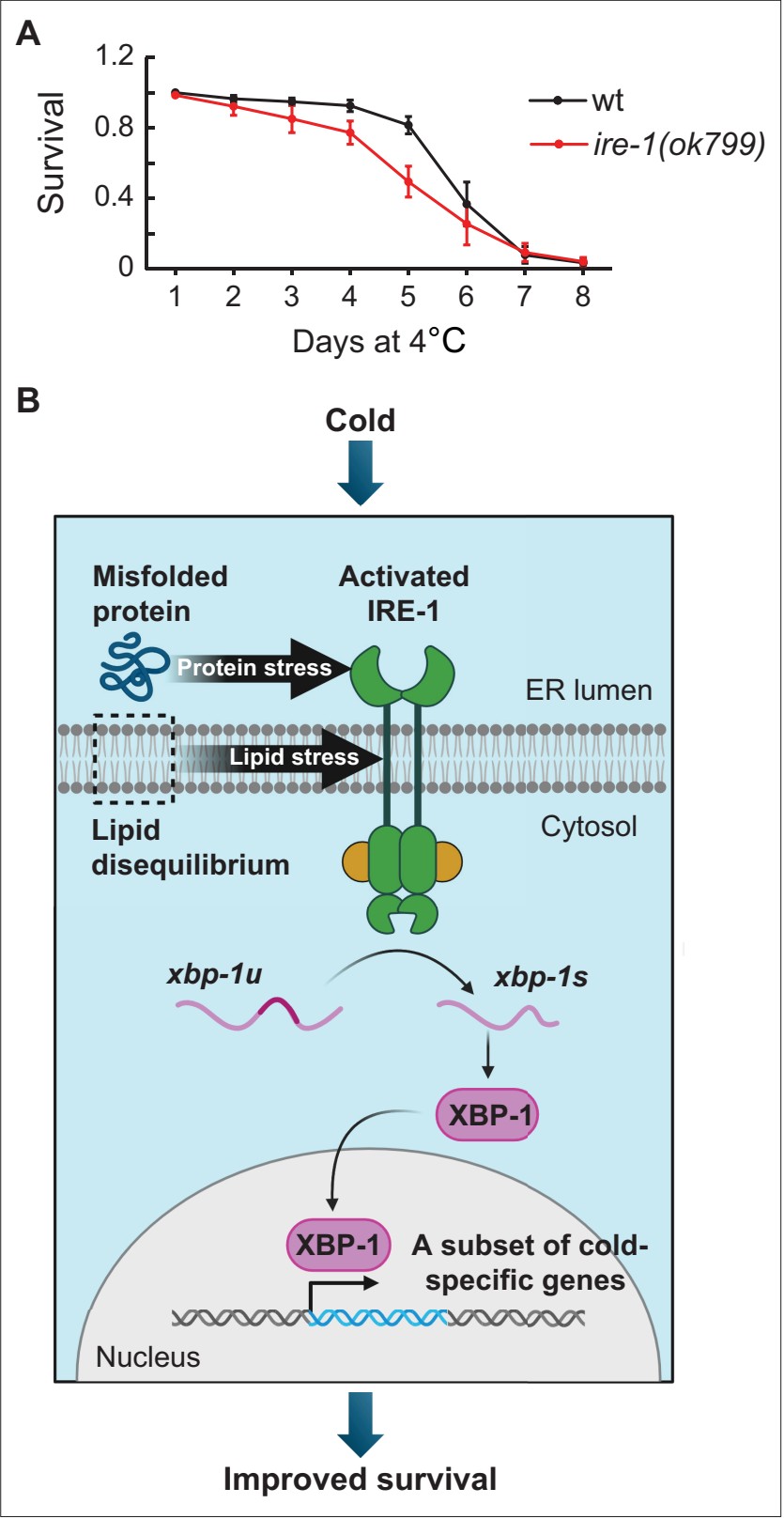

**Figure 7.** The IRE-1 pathway facilitates *C. elegans* survival during cold dormancy. (**A**) Survival of wild-type and *ire-1(ok799)* animals at 4 °C. Error bars indicate the SEM from six biological replicates. A minimum of 800 animals were scored per time point. Wilcoxon signed-rank test was used for statistical analysis; p=0.03. (**B**) A model for the IRE-1 function in hibernating nematodes. Cold exposure leads to increased protein misfolding and lipid disequilibrium

*Figure 7 continued on next page*

*Figure 7 continued*

in the endoplasmic reticulum (ER). These two stressors trigger the activation of IRE-1, which promotes the downstream processing of *xbp-1u* to *xbp-1s* mRNA. The functional XBP-1 transcription factor enters the nucleus to promote transcription of specific genes, including those facilitating cold survival. Created with BioRender.com.

The online version of this article includes the following figure supplement(s) for figure 7:

**Figure supplement 1.** Four-way Venn diagram depicting the overlap between genes with a minimum twofold enrichment (10°C to 4°C day 1) and previously reported IRE-1 responsive genes that are upregulated either during the UPR$^{PT}$ (*Shen et al., 2005*), upon shifting animals from 15°C to 2°C (*Dudkevich et al., 2022*), or during the UPR$^{LBS}$ (*Koh et al., 2018*).

possible explanation for these differences is that animals were grown at different starting temperatures (15 °C versus 20 °C) before cooling. *C. elegans* exhibits profound physiological differences between these two temperatures. For example, the above-mentioned demand for unsaturated FAs is heightened at 15 °C (*Svensk et al., 2013*), and the cold-sensitive TRPA-1 channel functions at 15° but not at 20 °C (*Xiao et al., 2013*). Additionally, while nematodes arrest development at 4 °C, they continue developing–albeit slowly–at 9 °C or higher (*Horikawa et al., 2024*). These observations suggest that *C. elegans* may activate IRE-1 through distinct mechanisms and with different outcomes depending on their physiological states and different cooling regimes.

What triggers IRE-1 activation in this study? Using the CPL-1$^{W32A, Y35A}$ protein folding sensor, we observed transient protein misfolding on day 1 of cold exposure, but not on day 3. Since protein synthesis is strongly reduced in the cold, the burden of misfolded proteins may similarly decrease over time, suggesting that additional cues drive UPR$^{ER}$ activation at later stages of hibernation. Our finding that supplementing unsaturated FAs did not prevent IRE-1 activation in hibernating nematodes aligns with previous research showing that inhibiting *C. elegans* fatty acid desaturases has little impact on their ability to survive severe cold (*Murray et al., 2007*). Combined with our observation that some FA desaturases may undergo additional translational repression in the cold, these results suggest that desaturated FAs are not limiting during *C. elegans* cold dormancy.

In contrast, choline supplementation reduced IRE-1-dependent expression during both early and later stages of hibernation, suggesting that changes in one or more choline derivatives contribute to ER stress. Since choline is essential for PC synthesis, a major component of biological membranes (*Kent, 1990*), the IRE-1 pathway may monitor a PC-sensitive aspect of ER biology. For example, it could detect alterations in transmembrane channels or peripheral membrane-binding proteins, whose activities depend on the physical properties of membrane lipids (*Allende et al., 2004*; *Janmey and Kinnunen, 2006*). However, choline also plays other roles, including in neurotransmitter synthesis and methylation metabolism. Thus, we cannot yet rule out the possibility that the protective effects of choline supplementation stem from functions outside PC synthesis.

Regardless of the exact role of IRE-1 signaling in *C. elegans* cooling, there is evidence suggesting that its connection to cold extends beyond nematodes. In cultured human neurons, moderate hypothermia activates the ER stress response and induces all three branches of the UPR, including IRE1 (*Rzechorzek et al., 2016*). The same study suggests that cooling-induced UPR$^{ER}$ may be neuroprotective. Whether this response also occurs in vivo and under deep hypothermia remains to be determined. If so, its manipulation could lead to improved procedures in organ transplantation and emergency medicine, where deep cooling induces a poorly understood state of preservation supporting vital organ functions of trauma patients (*Kutcher et al., 2016*).

## Methods
### *C. elegans* strains and maintenance

Unless stated otherwise, animals were maintained as previously described, grown at 20 °C on 2% Nematode Growth Media (NGM) agar plates, and seeded with the *E. coli* OP50 bacteria. All strains used in this study are listed in *Supplementary file 6*. Synchronized animals were obtained by extracting embryos from gravid adults with a bleaching solution (30% (v/v) sodium hypochlorite (5% chlorine) reagent (ThermoFisher Scientific; 419550010), 750 mM KOH). The embryos were left to hatch in the absence of food into arrested L1 larvae by overnight incubation at room temperature in M9 buffer (42 mM Na$_2$HPO$_4$, 22 mM KH$_2$PO$_4$, 86 mM NaCl, 1 mM MgSO$_4$).

## Cooling procedure

The cooling procedure was as previously published (*Habacher et al., 2016*; *Pekec et al., 2022*). In short, synchronized L1 larvae were grown at 20 °C until becoming young adults. The animals were then adapted to the cold for 2 hr at 10 °C, before being transferred to 4 °C.

## RNA-seq, polysome, and ribosome profiling

Animals were pre-grown at 20 °C, then moved to 10° for 2 hr, and then incubated at 4 °C. The reference 20 °C animals were collected at the time when others were moved to 10 °C (day 0). The RNA-seq, polysome profiling, and ribosome profiling were performed from two biological replicates as previously described (*Arnold et al., 2014*; *Scheckel et al., 2012*).

## Computational processing of RNA-seq and ribosome profiling

Both RNA-seq and ribosome profiling were processed with the ribo-seq pipeline in ORFik (v1.22.2) (*Tjeldnes et al., 2021*). The code for this pipeline can be found at https://www.bioconductor.org/packages/release/bioc/vignettes/ORFik/inst/doc/Ribo-seq_pipeline.html. This code was used with minor alterations (species, names of files, samples, etc.). The pipeline was run with default parameters indicated in the code above. It wraps several steps: stripping adapters using fastp (v0.23.4) with adapter sequence 'TGGAATTCTCGGGTGCCAAGG,' filtering contaminants (e.g. rRNA) and mapping reads with STAR (v2.7.11b) to the WBcel235 assembly and estimating expression (RPKM) using DESeq2(v1.42.1). Tracks used to display coverage for *Figure 3—figure supplement 2B* were created from the mapped BAM files using igvtools (v2.14.0) and the coverage was normalized to library size.

## Gene set enrichment analysis

Gene set enrichment was performed on the RNA-seq expression of all genes using the R Bioconductor package clusterProfiler (v4.10.1) and org.Ce.eg.db (v3.18.0). The top 10 categories from both the enriched and suppressed sets were visualized using enrichPlot (v1.22.0).

## Western blot analysis

Animals were harvested from plates, washed thrice in M9 buffer, and pelleted before being snap-frozen in liquid nitrogen. Protein extracts were prepared by grinding the pellet with a mortar and pestle in the presence of liquid nitrogen and dissolving in Lysis Buffer (50 mM HEPES (pH 7.4), 150 mM KCl, 5 mM $MgCl_2$, 0.1% Triton X-100, 5% glycerol (w/vol), 1 mM PMSF, 7 mg/ml cOmplete Proteinase Inhibitor Tablets (Roche, 11697498001)). Debris were removed by centrifugation at 16,100 x g for 20 min at 4 °C. Protein concentrations were measured by Bradford Assay (Bio-Rad). The required amount of 4 x NuPAGE LDS Sample Buffer (Invitrogen, NP0007) and 10 x NuPAGE Sample Reducing Agent (Invitrogen, NP0004) was added to the protein samples, followed by an incubation at 70 °C for 10 min. Proteins were separated by SDS-PAGE and transferred onto a polyvinylidene difluoride membrane by wet transfer. Membranes were washed thrice for 5 min with PBS-T, blocked for 1 hr in Intercept (TBS) Blocking Buffer (LI-COR, 927–60001), and incubated overnight at 4 °C with primary antibodies diluted in the same blocking buffer. The following primary antibodies were used: 1:10,000 monoclonal mouse anti-puromycin (Merck, MABE343), and 1:5000 polyclonal rabbit anti-actin (Abcam, ab8227). Detection was carried out with IRDye 680RD-conjugated goat anti-mouse secondary antibody (LI-COR Biosciences, 926–68070) or IRDye 800CW-conjugated goat anti-rabbit secondary antibody (LI-COR Biosciences, 926–32211) and infrared imaging (LI-COR Biosciences, Odyssey CLx).

## SUnSET assay

A total of 12,000 synchronized L1 larvae were grown at 20 °C and then cooled as described earlier or kept continuously at 20 °C. The animals were kept on multiple large 15 $cm^2$ plates for each sample to obtain sufficient material without crowding. The SUnSET assay was carried out essentially as previously described (*Arnold et al., 2014*), with modifications to assay protein synthesis at 4 °C. In brief, animals were washed twice in S-basal, resuspended in 4 ml S-medium, and transferred to a 50 ml Erlenmeyer. An overnight culture of *E. coli* OP50 was 10 x concentrated in S-medium and 750 µl was added to the animals together with 250 µl of 10 mg/ml puromycin (Millipore Sigma, P8833). The animals were grown for 4 hr at 200 rpm before harvesting. Animals were washed thrice with S-basal, pelleted, and snap-frozen in liquid nitrogen. Lysates were prepared as described in the western blot procedures and

40 µg of total protein was loaded per well. The incorporation of puromycin into nascent peptides was measured by normalizing band intensities from anti-puromycin to anti-actin antibodies. Three biological replicates were used for the final quantification.

## Construction of the *lips-11::GFP* reporter strain

The P*lips-11::GFP::unc-54* 3'UTR construct was generated via the MultiSite Gateway Technology (Thermo Fisher Scientific, 12537–023). The *lips-11* promoter (1140 bp) and GFP (867 bp) were amplified from *C. elegans* genomic DNA and a plasmid carrying GFP::H2B (pCM1.35) (*Merritt et al., 2008*), before being inserted into the entry vectors pDONRP4P1R and pDONR221, respectively (oligos are listed in *Supplementary file 6*). The resulting entry vectors were recombined along with the entry vector pCM5.37, carrying the *unc-54* 3'UTR (699 bp) (*Merritt et al., 2008*), and the destination vector pCFJ150 (*Frøkjaer-Jensen et al., 2008*), carrying chromosome II integration sites together with the *unc-119* (+) gene, resulting in the expression clone P*lips-11::GFP::unc-54* 3'UTR. Transgenic animals were obtained via single-copy integration into the *ttTi5605* locus on chromosome II by injecting EG4322 (*ttTi5605 II; unc-119(ed3) III*) animals with the expression clone (*Frøkjaer-Jensen et al., 2008*).

## Microscopy

Immediately prior to imaging, animals were anesthetized in a drop of 5 mM levamisole in M9 buffer on a 2% (w/v) agarose pad, clustered, covered with a cover slip, and immediately imaged with the Zeiss AxioImager Z1 microscope. Micrographs were acquired with an Axiocam MRm REV2 CCD camera using the Zen software (Zeiss) and processed with Image J. The specific area that was analyzed for fluorescent intensities, as well as the number of measurements, is indicated in each figure legend.

## Induction of ER stress by tunicamycin treatment

Animals were grown at 20 °C on NGM agar plates seeded with *E. coli* OP50. At the 2-d-old adult stage, animals were shifted to plates containing 20 µg/ml tunicamycin (Millipore Sigma, T7765), prepared from a 1 mg/ml stock solution dissolved in DMSO. Animals were then grown for 6 hr at 20 °C in the presence of tunicamycin. Control animals were treated similarly on plates containing the same volume of DMSO.

## Generation of iOP50 RNAi bacteria

Plasmids targeting *ire-1*, *xbp-1*, *pek-1*, or *atf-6* were extracted from overnight cultures of *E. coli* HT115 bacteria derived from the Ahringer library using the QIAprep Spin Miniprep Kit (QIAGEN, 27104) (*Kamath et al., 2003*). *E. coli* OP50 bacteria were rendered RNAi competent and chemically competent as previously described (*Neve et al., 2020*). The resulting iOP50 bacteria were transfected with 1 µl of the purified plasmids derived from the Ahringer library for downstream gene-specific RNAi or with the plasmid vector L4440 for mock RNAi (*Fire et al., 1998*; *Kamath et al., 2003*).

## RNAi

Gene-specific knockdown was achieved by feeding the animals with bacteria carrying plasmids expressing double-stranded RNA, sourced from either the Vidal or Ahringer libraries (*Fraser et al., 2000*; *Kamath et al., 2003*; *Rual et al., 2004*). Overnight cultures of *E. coli* HT115 (for RNAi at 20 °C) or *E. coli* iOP50 bacteria (for RNAi at 4 °C) were induced for 1 hr with 1 mM IPTG and seeded on NGM agar plates containing 1 mM IPTG and 50 µg/ml carbenicillin. Plates were additionally supplemented with fatty acids as described below for experiments combining RNAi-mediated gene knockdown and dietary supplementation. Synchronized L1 larvae were grown on the RNAi-inducing agar plates at 20 °C until reaching the 1-d-old adult stage and were either adapted to the cold as described and kept for 1 or 3 d at 4 °C or kept at 20 °C for the same duration. Bacteria containing the 'empty' L4440 vector were utilized as a mock RNAi control for all experiments.

## Dietary supplementation

For choline supplementation assays, a working stock of 200 mg/ml choline (Millipore Sigma, C7527) in water was added to autoclaved NGM media at 55 °C to a final concentration of 50 mM, except in the cold survival assay where the final concentration was 25 mM. Fatty acid-supplemented plates were prepared as previously described with adaptations (*Deline et al., 2013*). In brief, working stocks

of 100 mM were prepared by dissolving palmitic acid (Millipore Sigma, P9767), oleic acid (Millipore Sigma, O7501), and linoleic acid (Millipore Sigma, L8134) in 50% ethanol. Fatty acid sodium salts were added to an autoclaved NGM medium containing 0.1% Tergitol (NP40) to a final concentration of 0.8 mM. Fatty acid supplemented plates were covered with foil to prevent light oxidation. All plates were seeded with an overnight culture of *E. coli* OP50 and dried for 3 d before use.

## The assay for *C. elegans* cold survival

Cold survival experiments were performed as previously described (*Habacher et al., 2016*; *Pekec et al., 2022*). In brief, a minimum of 150 synchronized L1 larvae were grown and adapted to the cold as previously described for each time point. Animals were sampled at the indicated time points and their survival was scored after 24 hr recovery at 20 °C; those animals that were unresponsive to touch were considered dead. Cold survival was assessed from a minimum of three independent biological replicates, with a minimum of 200 animals used to assess the viability at each indicated time point.

## RT-qPCR

Approximately 6,000 animals were subjected to RNA extraction as previously described (*Arnold et al., 2014*), animals were either collected immediately before cold adaptation (20 °C day 0) or were adapted to the cold and kept for 1 or 3 d at 4 °C or kept at 20 °C for the same time prior to collection. Subsequently, genomic DNA was removed by DNase treatment and the quality of the RNA was assessed using the NanoDrop Spectrophotometer. Reverse transcription was performed by using the SuperScript IV First–Strand Synthesis System with random primers, following the protocol from the suppliers. RT-qPCR was performed with 2.5 µl of 1:5 diluted cDNA, 0.25 µl of 10 µM gene-specific primers (*Supplementary file 6*), and 2 µl of the HOT FIREPol EvaGreen qPCR Mix (Solis BioDyne, 08-36-00001) in a LightCycler 96 qPCR machine.

## Acknowledgements

We thank Dimos Gaidatzis and the Functional Genomics and Computational Biology facilities at the Friedrich Miescher Institute for Biomedical Research for the initial genomic analysis. We also thank Agnieszka Chabowska-Kita and the Laboratory of Animal Model Organisms (Institute of Bioorganic Chemistry PAS) for constructing the *lips-11* reporter strain, and Solfrid Lindhjem Kvinnesland (Department of Biosciences, UiO) for analyzing cold survival upon dietary choline supplementation. The research leading to these results received funding from the Norwegian Financial Mechanism 2014–2021 operated by the Polish National Science Center under the project contract nr UMO-2019/34 /H/ NZ3/00691. Some of the strains were provided by the CGC, which is funded by the NIH Office of Research Infrastructure Programs (P40 OD010440).

## Additional information

### Funding

| Funder | Grant reference number | Author |
| --- | --- | --- |
| The Norwegian Financial Mechanism 2014-2021 operated by the Polish National Science Center | UMO-2019/34/H/ NZ3/00691 | Rafal Ciosk |
| National Institutes of Health | Office of Research Infrastructure Programs P40 OD010440 | |

The funders had no role in study design, data collection and interpretation, or the decision to submit the work for publication.

### Author contributions

Melanie Lianne Engelfriet, Formal analysis, Validation, Investigation, Visualization, Methodology, Writing – original draft, Writing – review and editing; Yanwu Guo, Conceptualization, Formal analysis,

Investigation, Methodology; Andreas Arnold, Formal analysis, Validation, Investigation, Methodology; Eivind Valen, Formal analysis, Visualization, Methodology; Rafal Ciosk, Conceptualization, Resources, Supervision, Funding acquisition, Visualization, Methodology, Writing – original draft, Project administration, Writing – review and editing

**Author ORCIDs**
Melanie Lianne Engelfriet ⓘ https://orcid.org/0000-0002-0209-9492
Eivind Valen ⓘ https://orcid.org/0000-0003-1840-6108
Rafal Ciosk ⓘ https://orcid.org/0000-0003-2234-6216

Reviewer #2 (Public review): https://doi.org/10.7554/eLife.101186.3.sa1
Reviewer #3 (Public review): https://doi.org/10.7554/eLife.101186.3.sa2
Author response https://doi.org/10.7554/eLife.101186.3.sa3

---

## Additional files

### Supplementary files

Supplementary file 1. Total RNA-seq data of wild-type animals kept at the different indicated temperatures.

Supplementary file 2. Ribo-seq data of wild-type animals kept at the different indicated temperatures.

Supplementary file 3. Genes highlighted in *Figure 2A* in red.

Supplementary file 4. IRE-1 responsive genes from published datasets were used to generate the Venn diagram in *Figure 7—figure supplement 1*. The IRE-1 dependent cold genes were selected from Table S2 and Table S3 (*Dudkevich et al., 2022*): the genes were selected if upregulated in wild-type animals but not *ire-1(ok799)* mutants at 2 °C.

Supplementary file 5. Shared genes from the Venn diagram in *Figure 7—figure supplement 1*.

Supplementary file 6. The *C. elegans* strains (A) and oligos (B) were used in this study.

MDAR checklist

### Data availability

RNA sequencing information from this research can be found in the GEO repository under the following accession numbers: GSE269587 (RNA-seq) and GSE269589 (Ribo-seq).

The following datasets were generated:

| Author(s) | Year | Dataset title | Dataset URL | Database and Identifier |
|---|---|---|---|---|
| Engelfriet ML, Guo Y, Arnold A, Valen E, Ciosk R | 2024 | Remodeling of gene expression during *C. elegans* hibernation. [RNA-Seq] | https://www.ncbi.nlm.nih.gov/geo/query/acc.cgi?acc=GSE269587 | NCBI Gene Expression Omnibus, GSE269587 |
| Engelfriet ML, Guo Y, Arnold A, Valen E, Ciosk R | 2024 | Remodeling of gene expression during *C. elegans* hibernation. [Ribo-Seq] | https://www.ncbi.nlm.nih.gov/geo/query/acc.cgi?acc=GSE269589 | NCBI Gene Expression Omnibus, GSE269589 |

The following previously published datasets were used:

| Author(s) | Year | Dataset title | Dataset URL | Database and Identifier |
|---|---|---|---|---|
| Thibault G, Koh J, Wang L, Chabot C | 2018 | Lipid bilayer stress-activated IRE-1 modulates autophagy during endoplasmic reticulum stress | https://www.ncbi.nlm.nih.gov/geo/query/acc.cgi?acc=GSE99763 | NCBI Gene Expression Omnibus, GSE99763 |

*Continued on next page*

*Continued*

| Author(s) | Year | Dataset title | Dataset URL | Database and Identifier |
| --- | --- | --- | --- | --- |
| Dudkevich R, Henis-Korenblit S, Lebenthal-Loinger I | 2022 | Neuronal ire-1 coordinates an organism-wide cold stress response by regulating fat metabolism - Next Generation Sequencing Facilitates Quantitative Analysis of Wild Type and ire-1(ok799) Transcriptomes During Cold Stress | https://www.ncbi.nlm.nih.gov/geo/query/acc.cgi?acc=GSE193923 | NCBI Gene Expression Omnibus, GSE193923 |

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
