## [Editor Report · eLife Assessment]

This **useful** study reveals that as *C. elegans*, a poikilothermic ("cold-blooded") animal, adapt to cold (4ºC), they display a drastic reduction in translation (assessed by polysome profiling and SUNSET). The remaining translation (by ribo-seq) correlates with mRNA levels (by RNA-seq), and the changes in gene expression at least partially require IRE-1, an established endoplasmic reticulum stress sensor. The reviewers consider the data assessing global translation and RNA expression upon cold exposure and the data demonstrating the requirement of IRE-1 to be **solid**, but the conclusion that "transcription" is the major regulatory step and "lipid changes" can be a signal for IRE-1 activation in cold adapted worms needs substantially more evidence. Overall, this study demonstrated a good correlation between translation and RNA levels and yielded an inventory of gene changes as *C. elegans* adapt to cold, and will be of general interest to researchers interested in stress response and cold adaptation.

---

## [Referee Report · Reviewer #2 (Public review)]

Summary:

This study investigates cold induced states in *C. elegans*, using polysome profiling and RNA seq to identify genes that are differentially regulated and concluding that cold-specific gene regulation occurs at the transcriptional level. This study also includes analysis of one gene from the differentially regulated set, lips-11 (a lipase), and finds that it is regulated in response to a specific set of ER stress factors.

Strengths:

(1) Understanding how environmental conditions are linked to stress pathways is generally interesting.

(2) The study used well-established genetic tools to analyze ER stress pathways.

Weaknesses:

(1) The conclusions regarding a general transcriptional response are based on a few genes, with much of the emphasis on lips-11, which does not affect survival in response to cold.

(2) Definitive conclusions regarding transcription vs translational effects would require the use of blockers such as alpha-amanitin or cyclohexamide. Although this may be beyond the scope of the study, it does affect the breadth of the conclusions that can be made.

(3) Conclusions regarding the role of lipids are based on supplementation with oleic acid or choline, yet there is no lipid analysis of the cold animals, or after lips-1 knockdown. Although choline is important for PC production, adding choline in normal PC could have many other metabolic impacts and doesn't necessarily implicate PC without lipidomic or genetic evidence. Although they note the caveats, their evidence falls short of proving a role in PC production.

---

## [Referee Report · Reviewer #3 (Public review)]

Summary:

The authors sought to understand the molecular mechanisms that cells use to survive cold temperatures by studying gene expression regulation in response to cold in *C. elegans*. They determined whether gene expression changes during cold adaptation occur primarily at the transcriptional level and identified specific pathways, such as the unfolded protein response pathway, that are activated to possibly promote survival under cold conditions.

Strengths:

Effective use of bulk RNA sequencing (RNA-seq) to measure transcript abundance and ribosome profiling (ribo-seq) to assess translation rates, providing a comprehensive view of gene expression regulation during cold adaptation. This combined approach allows for correlation between mRNA levels and their translation, thereby offering evidence for the authors' conclusion that transcriptional regulation is the primary mechanism of cold-specific gene expression changes.

Weaknesses:

Many aspects of the weakness have been addressed by the revision. Still, the weak cold sensitivity phenotype observed in ire-1 mutants suggests the ER-UPR pathway's role is likely minor, modulatory or there is an unknown compensatory mechanism responsible for surviving cold.

---

## [Author Response]

The following is the authors’ response to the original reviews

**Public Reviews:**

**Reviewer #1:**
(…) some concerns with interpretations and technical issues make several major conclusions in this manuscript less rigorous, as explained in detail in comments below. In particular, the two major concerns I have: (1) the contradiction between the strong reduction of global translation, with puromycin incorporation gel showing no detectable protein synthesis in cold, and an apparently large fraction of transcripts whose abundance and translation in Fig. 2A are both strongly increased. (2) The fact that no transcripts were examined for dependance on IRE-1/XBP1 for their induction by cold, except for one transcriptional reporter, and some weaknesses (see below) in data showing activation of IRE-1/XBP-1 pathway. The conclusion for induction of UPR by cold via specific activation of IRE-1/XBP-1 pathway, in my opinion, requires additional experiments.

Relating to the first point, the results of puromycin incorporation and ribosome profiling are not contradictory. The former shows *absolute* changes in translation, i.e. changes in how much protein the cell is producing, while the latter shows relative changes between the produced proteins, i.e. how the cell prioritizes its protein production. An observed up-regulation in ribosome profiling does not necessarily mean (but could) that the corresponding protein goes up in absolute terms (units produced per time). Instead, it implies that out of the population of all translating ribosomes, a larger fraction is translating (prioritizing) this particular mRNA relative to other mRNAs. The second point is addressed later in the response.

Major concerns:(1) Fig. 1B shows polysomes still present on day 1 of 4ºC exposure, but the gel in Fig. 1C suggests a complete lack of protein synthesis. Why?

We realized that the selected gel exposure may give the false impression of a complete lack of puromycin incorporation at 4ºC. To avoid confusion, we now show in Figure 1 – figure supplement 1 the original gel image next to its longer exposure. The quantification of puromycin incorporation remains in Fig. 1C (it is based on 3 biological replicates and only one replicate is shown in the corresponding supplement). We hope it is now clear that there is an ongoing puromycin incorporation/translation at 4ºC, albeit much reduced compared with 20ºC.

What is then the evidence that ribosomal footprints used in much of the paper as evidence of ongoing active translation are from actual translating rather than still bound to transcripts but stationary ribosomes, considering that cooling to 4ºC is often used to 'freeze' protein complexes and prevent separation of their subunits? The authors should explain whether ribosome profiling as a measure of active translation has been evaluated specifically at 4ºC, or test this experimentally.

While the ribosomal profiling alone might not prove ongoing translation, the residual puromycin incorporation does (see the longer gel exposure in Figure 1 – figure supplement 1). To strengthen this argument, we selected two additional genes (*cebp-1* and *numr-1*) whose ribosomal footprints increase in the cold, and whose GFP-fusions were available from the CGC. Monitoring their expression, we observed the expected increase in the cold (see Figure 2 – figure supplement 3 A-B). The ongoing translation in the cold is also in line with our previous study (Peke et al., 2022), where we observed de novo protein synthesis of other proteins under the same cooling conditions as in this study.

They should also provide some evidence (like Western blots) of increases in protein levels for at least some of the strongly cold-upregulated transcripts, like lips-11.

As explained above, we addressed it by additionally examining two strains expressing GFP-fused proteins, whose translation in the cold is predicted to increase according to our ribosomal profiling data. See the new Figure 2 – figure supplement 3 A-B.

As puromycin incorporation seems to be the one direct measure of global protein synthesis here, it conflicts with much of the translation data, especially considering that quite a large fraction of transcripts have increased both mRNA levels and ribosome footprints, and thus presumably increased translation at 4ºC, in Fig. 2A.

We hope the above explanations put this concern to rest.

Also, it is not clear how quantitation in Fig. 1C relates to the gel shown, the quantitation seems to indicate about 50-60% reduction of the signal, while the gel shows no discernable signal.

A above, see a longer western blot exposure in Figure 1 – figure supplement 1 and note that the quantification is based on three biological replicates.

(2) It is striking that plips-11::GFP reporter is induced in day 1 of 4ºC exposure, apparently to the extent that is similar to its induction by a large dose of tunicamycin (Fig. 3 supplement),

We did not intend to compare the extend of induction between cold and tunicamycin treatment. The tunicamycin experiment was meant to confirm that, as suggested by expression data from Shen et al. 2005, *lips-11* is upregulated upon UPR activation.

…but the three IRE-1 dependent UPR transcripts from Shen 2005 list were not induced at all on day 1 (Fig. 4 supplement). Moreover, the accumulation of the misfolded CPL-1 reporter, that was interpreted as evidence that misfolding may be triggering UPR at 4ºC, was only observed on day 1, when the induction of the three IRE-1 targets is absent, but not on day 3, when it is stronger. How does this agree with the conclusion of UPR activation by cold via IRE-1/XBP-1 pathway?

In the originally submitted supplemental figure, we compared mRNA levels between day 1 animals at 20ºC versus 4ºC. However, as argued later by this reviewer, it may be better to use day 0 animals at 20ºC as the reference (since at 20ºC the animals will continue producing embryos). Thus, we repeated the RT-qPCR analysis with additional time points (and genes relevant to other comments). This analysis, now in Figure 4 – figure supplement 2, shows that these mRNAs (*dnj-27*, *srp-7*, and *C36B7.6*) increased already at day 1 in the cold compared with the reference 20ºC animals on day 0, and their levels increased further on day 3.

It is true that the authors do note very little overlap between IRE-1/XBP-1-dependent genes induced by different stress conditions, but for most of this paper, they draw parallels between tunicamycin-induced and cold induced IRE-1/XBP-1 activation.

We carefully re-examined the manuscript to ensure that we do not draw parallels between cold and tunicamycin treatment. The three genes (*dnj-27*, *srp-7*, and *C36B7.6*) were taken from Shen et al. because that study reported *lips-11* as an IRE-1-responsive gene, which we realized thanks to the Wormbase annotation of *lips-11*. Examining the three genes in our expression data, *srp-7* (like *lips-11*) is also upregulated more than 2-fold, while the other two genes go up but less than 2-fold. As mentioned by the reviewer, we note little overlap between the different stress conditions suggesting that the response is context dependent. Additional differences may arise if, as we hypothesize, UPR is activated in the cold in response to both protein and lipid stress. Note that the 2-fold cutoff used in the previous Figure 7 – figure supplement 1 was (erroneously) on the log2 scale, so showed genes upregulated at least 4-fold. We now corrected it to 2-fold. While there are now a few more overlapping genes, the overall conclusion, that there is little overlap between different conditions, did not change. We now list the shared genes in the new Supplementary file 5.

The conclusion that "the transcription of some cold-induced genes reflects the activation of unfolded protein response (UPR)..." is based on analysis of only one gene, lips-11. No other genes were examined for IRE-1 dependence of their induction by cold, neither the other 8 genes that are common between the cold-induced genes here and the ER stress/IRE-1- induced in Shen 2005 (Venn diagram in Figure 7 supplement), nor the hsp-4 reporter. What is the evidence that lips-11 is not the only gene whose induction by cold in this paper's dataset depends on IRE-1? This is a major weakness and needs to be addressed.Furthermore, whether induction by cold of lips-11 itself is due to IRE1 activation was not tested, only a partial decrease of reporter fluorescence by ire-1 RNAi is shown. A quantitative measure of the change of lips-11 transcript in ire-1 and xbp-1 mutants is needed to establish if it depends on IRE-1/XBP-1 pathway.

We now examined by RT-qPCR if the induction of the three genes from Shen at al. (*dnj-27*, *srp-7*, and *C36B7.6*), as well as *lips-11* and *hsp-4* depends on IRE-1. In the new Figure 4 – figure supplement 2, we show that the upregulation of all these genes is reduced in the cold in the *ire1* mutant (although in the wild type, the increase of *hsp-4* mRNA appeared to be non-significant, despite the observed upregulation of the *hsp-4* GFP reporter).

The authors could provide more information and the additional data for the transcripts upregulated by both ER stress and cold, including the endogenous lips-11 and hsp-4 transcripts: their identity, fold induction by both cold and ER stress, how their induction is ranked in the corresponding datasets (all of these are from existing data), and do they depend on IRE-1/XBP-1 for induction by cold?

As above, the dependence of endogenous *lips-11* and *hsp-4* on IRE-1 is now shown in the new Figure 4 – figure supplement 2, and the shared genes from Figure 7 – figure supplement 1 are listed in the new Supplementary file 5. We did not perform additional analysis comparing various data sets, as we felt that understanding the differences between IRE-1-mediated transcription outputs across different conditions goes well beyond this study.

Without these additional data and considering that the authors did not directly measure the splicing of xbp-1 transcript (see comment for Fig. 3 below), the conclusion that cold induces UPR by specific activation of IRE-1/XBP-1 pathway is premature.

To address the splicing of endogenous *xbp-1*, we examined our ribosome profiling data for the translation of spliced *xbp-1*, and found that the spliced variant is more abundant in the cold. This data is now shown in Figure 3 – figure supplement 2B.

There are also technical issues that are making it difficult to interpret some of the results, and missing controls that decrease the rigor of conclusions:(1) For RNAseq and ribosome occupancy, were the 20ºC day 1 adult animals collected at the same time as the other set was moved to 4ºC, or were they additionally grown at 20ºC for the same length of time as the 4ºC incubations, which would make them day 2 adults or older at the time of analysis? This information is only given for SUnSET: "animals were cultivated for 1 or 3 additional days at 4ºC or 20ºC".

In the RNAseq experiments, the 20ºC animals were collected at the same time as the others were moved to 10ºC (and then 4ºC), so they were *not* additionally grown at 20ºC. We make it now clear in Methods.

This could be a major concern in interpreting translation data: First, the inducibility of both UPR and HSR in worms is lost at exactly this transition, from day 1 to day 2 or 3 adults, depending on the reporting lab (for example Taylor and Dillin 2013, Labbadia and Morimoto, 2015, De-Souza et al 2022).

As explained above, the 20ºC animals were collected at the same time as the others were moved to 4ºC. Then, we reported before that ageing appears to be suppressed in animals incubated at 4ºC (Habacher et al., 2016; Figure S1C). Thus, it terms of their biological age, cold-incubated animals appear to be closer to the 20ºC animals at the time they are moved to the cold (day 0). Thus, the ageing-associated deterioration in UPR inducibility mentioned above presumably does not apply to cold-incubated animals, which is in line with the observed IRE-1-dependent upregulation of several genes in day 3 animals at 4ºC.

How do authors account for this? Would results with reporter induction, or induction of IRE-1 target genes in Fig. 4, change if day 1 adults were used for 20ºC?

Our analysis in Figure 4 – figure supplement 2 now includes 20ºC animals at day 0, 1, and 3.

Second, if animals at the time of shift to 4ºC were only beginning their reproduction, they will presumably not develop further during hibernation, while an additional day at 20ºC will bring them to the full reproductive capacity. Did 4ºC and 20ºC animals used for RNAseq and ribosome occupancy have similar numbers of embryos, and were the embryos at similar stages?

As explained above, the reference animals at 20ºC were young adults containing few embryos. Indeed, at 4ºC the animals do not accumulate embryos. Although we cannot say that for all genes, note that the genes analysed in Figure 4 – figure supplement 2 increase in abundance also when compared with the day 3 animals kept at 20ºC.

(2) Second, no population density is given for most of the experiments, despite the known strong effects of crowding (high pheromone) on *C. elegans* growth. From the only two specifics that are given, it seems that very different population sizes were used: for example, 150 L1s were used in survival assay, while 12,000 L1s in SUnSET. Have the authors compared results they got at high population densities with what would happen when animals are grown in uncrowded plates? At least a baseline comparison in the beginning should have been done.

None of the experiments involved crowded populations. In the SUnSET experiments, we just used larger and more plates to obtain sufficient material.

(3) Fig. 3: it is unclear why the accepted and well characterized quantitative measure of IRE1 activation, the splicing of xbp-1transcript, is not determined directly by RT-PCR. The fluorescent XBP-1spliced reporter, to my knowledge, has not been tested for its quantitative nature and thus its use here is insufficient. Furthermore, the image of this fluorescent reporter in Fig. 3b shows only one anterior-most row of cells of intestine, and quantitation was done with 2 to 5 nuclei per animal, while lips-11 is induced in entire intestine. Was there spliced XBP-1 in the rest of the intestinal nuclei? Could the authors show/quantify the entire animal (20 intestinal cells) rather than one or two rows of cells?

As explained above, we now included the analysis of *xbp-1* splicing in Figure 3 – figure supplement 2B. As for the fluorescent reporter, it is difficult to measure all gut nuclei since part of the gut is occluded by the gonad. Nonetheless, we do see induction of the reporter in other gut nuclei and show now additional examples from midgut in Figure 3 – figure supplement 2A.

(4) The differences in the outcomes from this study and the previous one (Dudkevich 2022) that used 15ºC to 2ºC cooling approach are puzzling, as they would suggest two quite different IRE-1 dependent programs of cold tolerance. It would be good if authors commented on overlapping/non-overlapping genes, and provided their thoughts on the origin of these differences considering the small difference in temperatures.

Indeed, there seem to be substantial differences between different temperatures and cooling paradigms. While understanding the *C. elegans* responses to cold is still in its infancy, one possible explanation for the observed differences is that we used different starting growth temperatures. While the initial populations in our study were grown at 20ºC, Dudkevich et al. used 15ºC. Worms display profound physiological differences between these two temperatures. For example, Xiao et al. (2013) showed that the cold-sensitive TRPA-1 channel is important at 15ºC but not 20ºC. Thus, the trajectories along which worms adapt to near freezing temperature may vary depending on their initial physiological state (and perhaps the target temperature, as we used 4ºC and they 2ºC). We now expanded argumentation on this topic in Discussion. I should also say that we planned on testing NLP-3 function in our paradigm, but our request for strains remained unanswered.

Second, have the authors performed a control where they reproduced the rescue by FA supplementation of poor survival of ire-1 mutants after the 15ºC to 2ºC shift? Without this or another positive control, and without measuring change in lipid composition in their own experiments, it is unclear whether the different outcomes with respect to FAs are due to a real difference in adaptive programs at these temperatures, or to failure in supplementation?

While we did not re-examine the findings by Dudkevich et al., we did include now another positive control. As reporter by Hou et al. (2014), supplementing unsaturated FAs rescues the induction of the *hsp-4* reporter in *fat-6* RNAi-ed animals. Although we were able to reproduce that result (Figure 6 – figure supplement 1), the same supplementation procedure did not suppress the *lips11* reporter (Figure 6 – figure supplement 2).

(5) Have the authors tested whether and by how much ire-1(ok799) mutation shortens the lifespan at 20ºC? This needs to be done before the defect in survival of ire-1 mutants in Fig. 7a can be interpreted.

The lifespan at standard cultivation temperature was examined by others (Henis-Korenblit et al., 2010; Hourihan et al., 2016), showing that *ire-1(ok799)* mutants live shorter. However, while some mechanism that prolong lifespan may also improve cold survival, the two phenomena are not identical and whether IRE-1 facilitates longevity and cold survival in the same or different way remains to be seen.

**Reviewer #2**:(1) The conclusions regarding a general transcriptional response are based on one gene, lips-11, which does not affect survival in response to cold. We would suggest altering the title, to replace "Reprograming gene expression: with" Regulation of the lipase lips-11".

We now examined IRE-1 dependent induction of additional genes – see Figure 4 – figure supplement 2. While we do not know what fraction of cold-induced genes depends on IRE-1, we feel that our findings justify the statement that that gene expression in the cold *involves* the IRE1/XBP-1 pathway (title) or that that the transcription of *some/a subset of* cold-induced genes depend on this pathway (in abstract, model, and discussion).

(2) There is no gene ontology with the gene expression data.

We now included the top 10 most enriched and suppressed gene categories between 10ºC and 4ºC (since the biggest change happens between these conditions, as shown in Figure 2 – figure supplement 1A). This is now included in the Figure 2 – figure supplement 2.

(3) Definitive conclusions regarding transcription vs translational effects would require use of blockers such as alpha amanatin or cyclohexamide.

As explained also for reviewer 1, we confirmed now that at least some genes, whose translation is upregulated based on the ribosome profiling, are indeed upregulated in the cold at the protein level (Figure 2 – figure supplement 3A-B). Thus, the increase in ribosomal occupancy seems to accurately reflect increased translation. Since mRNA levels correlate overall with the ribosomal occupancy, it appears that the mRNA levels are the main determinants of the translation output. Because the *lips-11* promoter is sufficient to upregulate the GFP reporter in the cold, it further suggests that the regulation happens at the transcription level. It is true that at this point we cannot completely rule out the effects of mRNA stability, which we clearly acknowledge in the discussion.

(4) Conclusions regarding the role of lipids are based on supplementation with oleic acid or choline, yet there is no lipid analysis of the cold animals, or after lips-1 knockdown.

We agree that this is an important direction for future studies but feel that lipidomic analysis goes beyond the scope of current work.

Although choline is important for PC production, adding choline in normal PC could have many other metabolic impacts and doesn't necessarily implicate PC without lipidomic or genetic evidence.

We agree and acknowledge it now in Discussion: “However, choline also plays other roles, including in neurotransmitter synthesis and methylation metabolism. Thus, we cannot yet rule out the possibility that the protective effects of choline supplementation stem from functions outside PC synthesis.”

**Reviewer #3:**
The study has several weaknesses: it provides limited novel insights into pathways mediating transcriptional regulation of cold-inducible genes, as IRE-1 and XBP-1are already well-known responders to endoplasmic reticulum stress, including that induced by cold.

We presume the reviewer refers to the study by Dudkevich et al. (2022). As explained in our manuscript, there are important differences between that study and ours in how the IRE-1 signalling is utilized and to what ends.

Additionally, the weak cold sensitivity phenotype observed in ire-1 mutants casts doubt on the pathway's key role in cold adaptation. The study also overlooks previous research (e.g.PMID: 27540856) that links IRE-1 to SKN-1, another major stress-responsive pathway, potentially missing important interactions and mechanisms involved in cold adaptation.

We state in the manuscript that the IRE-1 pathway plays a modest but significant role in cold adaptation and state in the Fig. 7 model and Discussion that additional pathways work alongside IRE-1 to drive cold-specific gene expression.

**Recommendations for the authors:**

**Reviewer #1:**
Minor comments:(1) Fig. 2B - reporter expression seems to be already present in the intestine of 20ºC animals. What is the turnover rate of GFP in the intestine and how is it affected by the temperature shift? If GFP degradation is inhibited, could it explain the increase in signal in 4ºC animals, rather than increased transcription? This seems to be true for the hsp-4 transcriptional reporter, as the GFP fluorescence appears to increase during 4ºC incubation (Fig. 4a), but the hsp-4 message levels are only increased after 1 day but not in later days at 4ºC, based on the RNAseq in provided dataset. How well do changes in lips-11 reporter fluorescence correspond to the changes in the endogenous lips-11 transcript?

Note that increased GFP fluorescence is accompanied by increased mRNA levels. In addition to the RNAseq data, we now also examined changes of the endogenous *lips-11* transcript by RTqPCR and observed its strong (and IRE-1 dependent) upregulation in the cold– see Figure 4 – figure supplement 2. Moreover, we now included two other examples of GFP-tagged proteins whose fluorescence increases in the cold, concomitant with increased mRNA levels and ribosomal occupancy (Figure 2 – figure supplement 2A-B).

(2) Descriptions of methods to measure different aspects of translation are very abbreviated and in some places make it difficult to understand the paper. One example - what is RFP in Fig. 2a?

We replaced now “RFP” with “RPF” (ribosome protected fragment) and the abbreviation is explained firsts time it is used.

(3) How was the effectiveness of RNAi at 4ºC validated?

As explained in Methods, we subjected animals to RNAi long before they were transferred to 4ºC, so the corresponding protein is depleted prior to cooling.

(4) Several of the conclusions on translation and ribosomal occupancy are written in a somewhat confusing way. For example, the authors state that "shift from 10ºC to 4ºC had a strong effect" when describing "impact on translation (ribosomal occupancy)" (page 4), but in the next sentence, they state "a good correlation between mRNA levels and translation (Figure 2A)". Was ribosomal occupancy normalized to the transcript abundance?

We do not perceive any discrepancy between the two statements. The former refers to the difference between time points, where we observed the largest change in both the transcriptome and ribosomal occupancy from 10ºC to 4ºC (as can be inferred in the PCA plot in Figure 2 - figure supplement 1). The latter refers to the observation that changes in mRNA levels mirrored, in most of cases, similar changes in the ribosomal occupancy.

The ribosomal occupancy was not normalized, as that would essentially normalize the y-axis (ribosomal occupancy) with the x-axis (mRNA), and so express changes in “translational efficiency” as a function of changes in mRNA abundance. While this type of analysis can also reveal interesting biological phenomena, it would explore a different question.

(5) "For most transcripts ... increased the abundance of a particular protein appears to correlate depend primarily on the abundance of its mRNA" (page 5). This is an overstatement, the protein levels were not quantified.

As explained above, we now additionally monitored the expression of two GFP-tagged proteins (CEBP-1 and NUMR-1). Monitoring their expression, we observed the expected increase in GFP fluorescence in the cold (see Figure 2 – figure supplement 3 A-B). While we did not examine them also by western blot, these observations are in line with our conclusions.

(6) The statement "Since transcription is the main determinant of mRNA levels, these results suggest that cold-specific gene expression primarily depends on transcription activation" seems to assume that message degradation doesn't have much of an impact at 4ºC. What is the evidence here? The authors themselves later suggest either transcription or mRNA stability in Discussion.

While we cannot exclude that mRNA stability of some genes may be affected, this concern is more valid for the messages that go down in the cold. Although we have done it for only selected genes, each time we observed an increase in the mRNA levels, we also observed the corresponding increase in the protein; this study and Pekec et al. (2022). Then, the *lips-11* reporter was designed to monitor the activity of its promoter, which we showed in sufficient to upregulate reporter GFP in the cold. We have now expanded the corresponding paragraph in Discussion, which will hopefully come across as more balanced.

**Reviewer #2:**
(1) Alter title, conclusions to better reflect specific nature of the work.

We now provided additional data and feel that it justifies our conclusions and title.

(2) Use Gene Ontology searches to look at patterns of gene expression in RNA seq data.

We now show it in Figure 2 – figure supplement 2.

(3) Use genetic or lipidomic tools rather than solely adding exogenous lipids.

We agree that lipidomic analysis is an important direction for future research, but feel that lipidomic analysis and further genetic experiments go beyond the scope of current manuscript.

**Reviewer #3:**
To strengthen the evidence for the role of IRE-1 in cold adaptation, the authors might consider performing additional functional assays, such as testing the effects of IRE-1 and XBP-1 mutations under varying cold conditions and testing the genetic interaction of ire-1 with xbp-1, skn-1, and hsf-1 in cold sensitivities. It is also worth using alternative approaches such as independent alleles of ire-1, knockdowns or tissue-specific knockouts (without potential developmental compensation in global constitutive mutants) to better characterize the contribution of IRE-1 to cold adaptation. Additionally, studies that examine tissue-specific responses to cold exposure could provide important insights, as different tissues may utilize distinct molecular pathways to adapt to cold stress.

We also tested *ire-1* and *xbp-1* functions by RNAi-mediated depletion. SKN-1 is a good candidate for future studies, but Horikawa at al. (2024) showed that HSF-1 is not required for cold dormancy (at 4ºC); we also show now that HSF-1::GFP does not increase in the cold (Figure 2 – figure supplement 3C).

This reviewer also recommends clarifying the novelty of your findings in the context of existing literature, particularly regarding the established roles of IRE-1 and XBP-1 in responding to endoplasmic reticulum stress.

The entry point of this study was to clarify a long-standing problem in hibernation research, i.e., the apparent discrepancy between a global translation repression and de novo gene expression observed in the cold. By connecting cold-mediated expression of some genes to the IRE-1/XBP1 pathway, we strengthen the argumentation for transcription-mediated gene regulation in hibernating animals. We did go the extra mile to test the possible reason behind the activation of UPR^ER^ in the cold but feel that a deeper analysis deserves a separate study.

The term "hibernation" should be avoided or reworded since the study does not provide direct behavioral or physiological evidence for hibernation-like states; instead, the manuscript could refer to "cold-induced responses" or "adaptations to cold temperatures."

The term “hibernation” was used before even in the context of the *C. elegans* dauer state, which, arguably, is even less appropriate. In addition to a global suppression of translation shown here, we reported before that the same cooling regime suppresses ageing (Habacher et al., 2016; Figure S1C). Incubating at 4ºC also arrests *C. elegans* development (Horikawa et al., 2024). Thus, while the worm and mammalian hibernation are certainly not equivalent – which we clearly spell out – we like to use “hibernation” interchangeably with “cold dormancy” to draw attention to a fascinating aspect of *C. elegans* biology. Still, we use now quotation marks in the title to avoid misunderstanding.

The discussion could be strengthened by addressing the relevance of prior studies, such as those linking IRE-1 to SKN-1 (PMID: 27540856), TRPA-1 (PMID: 23415228), ZIP-10 (PMID: 29664006), HSF-1 (PMID: 38987256) in cold adaptation and elaborating on how your findings provide new

The IRE-1/SKN-1 and ZIP-10 papers are now mentioned when describing the model in Figure 7. The TRP-1 and HSF-1 papers are cited when discussing physiological differences between different cold temperatures. Consistent with our studies, the HSF-1 paper shows that nematodes enter a dormant state at 4ºC (but at 9ºC and higher temperatures continue developing). Importantly, HSF-1 promotes the development at 9ºC but is not important for the arrest at 4ºC. We also shown now in Figure 2 – figure supplement 3C that HSF-1 does not go up at 4ºC.